# Causal Canonical Modeling for Confounding Robust Treatment Evaluation

## Abstract

The existence of confounding bias presents one of the central challenges in policy evaluation, since the target effects of actions are not identifiable (i.e., underdetermined) from observational data. This paper investigates treatment/causal effect evaluation over the continuous action-reward domain from confounded observations, while requiring only basic temporal ordering between the treatment and the outcome, and Lebesgue integrability over the target treatment effect. We introduce a novel family of causal canonical models that can effectively approximate the observational and interventional distributions of any causal model consisting of continuous action and reward variables. Building on this newfound universal approximation property, we develop a novel family of generative models via a mixture of Gaussian processes that allow one to derive posterior distributions over unknown causal effects provided with confounded observations.

## 1 Introduction

Evaluating the causal effects of how a treatment affects a primary outcome is of interest across many fields of science, including econometrics (Rosenbaum & Rubin, 1983), healthcare (Murphy, 2003), biostatistics (Wright, 1928), social sciences (Imbens & Rubin, 2015), and, recently, artificial intelligence Bareinboim & Pearl (2016). For example, policymakers may want to assess the impact of a training program on job employment; physicians often seek to understand the effectiveness of a drug in treating a disease; engineers might investigate the relationship between ad placement and the resulting click-through rates of consumers. When treatment is assigned randomly or the treatment allocation policy generating observations is fully known, the target causal effects are recoverable from the observed data in a straightforward fashion. On the other hand, in many practical applications, the learner does not know or control the treatment assignment mechanism. This gives rise to confounding bias in the offline data, resulting in spurious correlations during the treatment evaluation (Pearl, 1995).

To address the challenges of confounding bias, researchers and practitioners may exploit additional theoretical assumptions about the underlying environment (Wright, 1928; Angrist et al., 1996). The problem of identifying causal effects from the combination of observational data and assumptions has been extensively studied under the rubrics of causal inference (Wright, 1928; Angrist et al., 1996; Pearl, 2000; Spirtes et al., 2001). Particularly, qualitative causal knowledge about the environment could be represented in the form of a *directed acyclic causal diagram* (Pearl, 2000, Ch. 1.2). Various criteria and algorithms have been developed based on the causal diagram (Pearl, 2000; Spirtes et al., 2001; Bareinboim & Pearl, 2016). This means the conditions under which the target effects are identifiable from data have been understood. For example, a criterion called *back-door* (Pearl, 2000, Ch. 3.2.2) allows one to identify causal effects by covariate adjustment. Efficient estimators were developed based on the inverse propensity score weighting (Rosenbaum & Rubin, 1983; Bang & Robins, 2005) and off-policy learning (Dudík et al., 2011; Li et al., 2015; Munos et al., 2016; Thomas & Brunskill, 2016). Additionally, one may incorporate parametric assumptions about the forms of underlying functions and distributions to facilitate identification. For example, algorithms exist to identify causal effects from confounded observations, provided with the linearity assumption (Wright, 1934; Chen et al., 2017). For non-linear systems, parametric conditions have been proposed under which the target effect is identifiable (Wang & Blei, 2019; Manski & Pepper, 1998; Maiti et al., 2025).

However, in many practical applications, the combination of qualitative knowledge and observed data does not always allow one to uniquely compute the target effect. Such challenging cases are referred to as *non-identifiable* (Pearl, 2000, Def. 3.2.2). The following example illustrates such challenges.

**Example 1.** Consider a data-generating process concerning a system with an action $X$ and a reward $Y$, values of which are decided by functions $X \leftarrow -U/2$ and $Y \leftarrow -X^2 + U^2$, respectively; $U$ is an unobserved variable drawn from normal distributions with mean $\mu = 0$ and variances $\sigma^2 = 1$. Fig. 1 shows the observed samples (highlighted in blue) generated by this system, summarized as the observational distribution $P(X, Y)$. We also show samples (orange) collected by randomly assigning action values over a real interval $[-2, 2]$, summarized as an interventional distribution $P_x(Y)$. Interestingly, the observational distribution $P(X, Y)$ deviates significantly from the interventional distribution $P_x(Y)$. This is due to the unobserved confounder $U$, which introduces a spurious correlation between treatment $X$ and outcome $Y$, making some actions appear more effective.

It has been acknowledged in the literature that treatment effects $\mathbb{E}_x[Y]$ are not identifiable in such systems (Huang & Valtorta, 2006; Shpitser & Pearl, 2006). Particularly, one could construct an alternative system with reward function $Y \leftarrow 3/4 \times U^2$ that generates the same observed data, but gives a different evaluation on the treatment effects $\mathbb{E}_x[Y]$. To witness, we apply Gaussian process regression to the observed samples. The learned function, shown in Fig. 1, perfectly fits the conditional reward $\mathbb{E}[Y \mid x]$. However, it fails to generalize to the actual treatment effect $\mathbb{E}_x[Y]$ when one actively intervenes in the system by setting the action $X$ to a constant $x \in [-2, 2]$. ∎

More recently, there has been an increasing body of work studying the approximation property of generative models (Zhang & Bareinboim, 2021; Zhang et al., 2022; Xia et al., 2021; 2022; Nasr-Esfahany et al., 2023) in evaluating causal effects, particularly in non-identifiable settings. These novel approximation properties permit one to infer the parametric forms of the underlying functions and latent confounders (unknown and untestable) from weak parametric knowledge about the observed domains (valid and testable). For instance, for an unknown causal model with discrete observed variables, (Zhang et al., 2022) showed that the domain of latent confounders could be discretized without loss of generality. Identifying unknown causal effects in this class

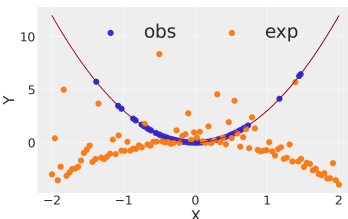

Figure 1: Samples drawn from the observational $P(X, Y)$ (blue) and interventional $P_x(Y)$ (orange) distributions.

of discrete generative models is reducible to solving a series of polynomial programs (Zaffalon et al., 2020), which could be further simplified to linear programs in specific settings (Balke & Pearl, 1997; Joshi et al., 2024). However, challenges still exist in applying causal generative modeling to observed data with complex and possibly continuous domains.

The goal of this paper is to address these challenges by proposing novel causal generative models that can approximate observational and interventional distributions in any unknown causal model with continuous treatment and outcome variables. This newfound approximation property enables us to develop robust partial identification algorithms for inferring unknown causal effects from confounded observational data in continuous domains. More specifically, our contributions are summarized as follows. (1) We introduce a novel family of *causal canonical models* with discrete latent states and continuous functional mapping among variables in the system. (2) We formally show that the proposed model class could approximate the observational and interventional distributions in any causal model with continuous treatment and outcome with arbitrary accuracy. (3) We reparameterize this canonical representation into a novel family of generative models consisting of mixtures of Gaussian processes. Inferencing in this generative model yields a posterior distribution over parameters of the target causal effects. For the sake of the space constraints, all the proofs are provided in Sec. A; details about the experimental setup are provided in Sec. B.

## 2 TREATMENT EVALUATION FROM CONFOUNDED OBSERVATIONS

This section introduces some basic notations and definitions that will be used throughout the paper. We use capital letters to denote variables ($X$) and small letters for their values ($x$). For an arbitrary set $\boldsymbol{X}$, let $|\boldsymbol{X}|$ be its cardinality. $P(\boldsymbol{X})$ denote the probability distribution on variables $\boldsymbol{X}$. We consistently use $P(\boldsymbol{x})$ as a shorthand for probability $P(\boldsymbol{X} = \boldsymbol{x})$; similarly, $P(\mathbb{X})$ stands for probability $P(\boldsymbol{X} \in \mathbb{X})$ of the event where $\boldsymbol{X}$ is contained in a collection $\mathbb{X}$ of possible realizations $\boldsymbol{X} = \boldsymbol{x}$.

**Structural Causal Models.** We will focus on the structural causal models (Pearl, 2000; Bareinboim et al., 2020) graphically described in Fig. 2a where $\boldsymbol{X}$ is is a treatment/action; $Y$ is the outcome/reward; and $\boldsymbol{U}$ are unobserved exogenous variables representing uncertainties in the environment. For an arbitrary causal model $\mathcal{M}$, the values of action $\boldsymbol{X}$ are decided by a function $\boldsymbol{X} \leftarrow f_X(\boldsymbol{U})$; and the values of reward $Y$ are decided by a function $Y \leftarrow f_Y(\boldsymbol{X}, \boldsymbol{U})$. The causal mechanisms generating latent variables $\boldsymbol{U}$ are not explicitly described. Instead, values of $\boldsymbol{U}$ are drawn from an exogenous distribution $P(\boldsymbol{U})$ over the $m$-dimensional real space $\mathbb{R}^m$. An agent passively observes the environment and receives action-reward pairs $(\boldsymbol{x}, y)$. The probabilities of occurrence of observed events are summarized as the *observational distribution* $P(\boldsymbol{X}, Y) \triangleq P(\boldsymbol{X}, Y; \mathcal{M})$.

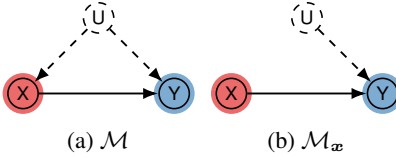

An intervention on action $\boldsymbol{X}$, denoted by $\mathrm{do}(\boldsymbol{X} \leftarrow \boldsymbol{x})$ (for short, $\mathrm{do}(\boldsymbol{x})$), is an operation where values of $\boldsymbol{X}$ are set to constants $\boldsymbol{x}$, replacing the function $f_X$ that would normally determine the action. For a causal model $\mathcal{M}$, let $\mathcal{M}_{\boldsymbol{x}}$ be a submodel of $\mathcal{M}$ induced by intervention $\mathrm{do}(\boldsymbol{x})$. Given unit $\boldsymbol{U} = \boldsymbol{u}$, *potential outcome* $Y$ to action $\mathrm{do}(\boldsymbol{x})$, denoted as $Y_{\boldsymbol{x}}(\boldsymbol{u})$, is the solution of outcome $Y$ in submodel $\mathcal{M}_{\boldsymbol{x}}$, i.e., $Y_{\boldsymbol{x}}(\boldsymbol{u}) \triangleq Y_{\mathcal{M}_{\boldsymbol{x}}}(\boldsymbol{u})$. The *interventional distribution* $P_{\boldsymbol{x}}(Y)$ induced by $\mathrm{do}(\boldsymbol{x})$ is defined as the joint distribution over the potential outcome, i.e.,

(a) $\mathcal{M}$      (b) $\mathcal{M}_{\boldsymbol{x}}$

Figure 2: Causal diagrams of (a) a bandit model with a treatment/action $\boldsymbol{X}$ and outcome/reward $Y$; (b) the submodel induced by intervention $\mathrm{do}(\boldsymbol{X} \leftarrow \boldsymbol{x})$.

$P_{\boldsymbol{x}}(Y) \triangleq P(Y_{\boldsymbol{x}}; \mathcal{M})$. The treatment effect of action $\boldsymbol{x}$ is thus defined as the expected value $\mathbb{E}_{\boldsymbol{x}}[Y]$.

**Treatment Evaluation Problem.** In this paper, our goal is to infer the treatment effect $\mathbb{E}_{\boldsymbol{x}}[Y]$ for all possible actions $\boldsymbol{x}$ (i.e., **output**) from the observed samples drawn from the distribution $P(\boldsymbol{X}, Y)$ (i.e., **input**). We will also make the following **assumptions**. First, the treatment $\boldsymbol{X}$ is a vector variable in an $n$-dimensional real space $\boldsymbol{X} \in \mathbb{R}^n$; the reward $Y$ is a continuous variable taking a real value in $Y \in \mathbb{R}$. For any action $\boldsymbol{x} \in \mathbb{R}^n$, the treatment effect $\mathbb{E}_{\boldsymbol{x}}[Y]$ is Lebesgue-integrable. That is, $\mathbb{E}_{\boldsymbol{x}}[Y] : \mathbb{R}^n \mapsto \mathbb{R}$ is a Lebesgue-measurable function satisfying $\int_{\mathbb{R}^n} |\mathbb{E}_{\boldsymbol{x}}[Y]| \, d\boldsymbol{x} < \infty$, which includes continuous functions and functions such as the sgn function. We will consistently use the $L_1$ norm $\|\boldsymbol{x}\|_1 = \sum_{i=1}^n |x_i|$ to measure approximation error.

## 3 APPROXIMATING STRUCTURAL CAUSAL MODELS

This section will introduce a canonical family of causal models that can effectively approximate the observed data and treatment effects in any structural causal model compatible with Fig. 2a. We will also provide tools and concepts explaining the intuitions behind the approximation procedure.

First, we will describe some necessary notation. We will consistently use $\mathbb{U}$ to denote a subset in the exogenous domain $\mathbb{R}^m$, and $\mathbb{X}$ stands for a subset contained in the action domain $\mathbb{R}^n$. Fix an arbitrary integer $N \in \mathbb{N}^+$. Let a sequence of constants $\boldsymbol{x}_1, \ldots, \boldsymbol{x}_N$ in the action domain $\mathbb{R}^n$, and let $\mathbb{U}_1, \ldots, \mathbb{U}_N$ be disjoint subsets in the exogenous domain $\mathbb{R}^m$. Similarly, let $y_1, \ldots, y_N$ be a finite sequence of constants in the reward domain $\mathbb{R}$, and let $\mathbb{X}_1, \ldots, \mathbb{X}_N$ be disjoint subsets in the action domain $\mathbb{R}^n$. Simple functions determining values of action $\boldsymbol{X}$ and reward $Y$ are given by

$$\widehat{f}_{\boldsymbol{X}}(\boldsymbol{u}) = \sum_{i=1}^N \boldsymbol{x}_i \mathbb{1}_{\mathbb{U}_i}(\boldsymbol{u}), \qquad\qquad \widehat{f}_Y(\boldsymbol{x}, \boldsymbol{u}) = \sum_{i=1}^N y_i \mathbb{1}_{\mathbb{X}_i}(\boldsymbol{x}) \mathbb{1}_{\mathbb{U}_i}(\boldsymbol{u}), \qquad (1)$$

where $\mathbb{1}_{\mathbb{U}}$ and $\mathbb{1}_{\mathbb{X}}$ are the indicator functions of the subset $\mathbb{U}_i \subseteq \mathbb{R}^m$ and $\mathbb{X}_i \subseteq \mathbb{R}^n$ respectively. A causal canonical model compatible with the qualitative assumptions of Fig. 2a is an SCM where simple functions determine the values of its observed variables. Formally,

**Definition 1.** A causal canonical model (CCM) $\mathcal{M}$ is an SCM where its action $\boldsymbol{X}$ and reward $Y$ are decided by simple functions $\widehat{f}_{\boldsymbol{X}}$ and $\widehat{f}_Y$ defined in Eq. (1).

Let $\mathscr{M}$ be the set of all SCMs compatible with the causal graph in Fig. 2a. Similarly, let $\mathscr{N}$ denote the space of CCMs compatible with Fig. 2a. By the definition of CCMs in Def. 1, the canonical space $\mathscr{N}$ must be strictly contained in the original space $\mathscr{M}$. We want to identify a subspace of causal models contained in $\mathscr{M}$ with the following *causal approximation property*, i.e.,

**Definition 2** (Causal Approximation Property). Let $\mathscr{M}$ be the set of all SCMs described in Fig. 2a. A subset $\mathscr{N} \subset \mathscr{M}$ is said to satisfy the *causal approximation property* if given any SCM $\mathcal{M} \in \mathscr{M}$

and any $\epsilon > 0$, there exists an alternative causal model $\mathcal{N} \in \mathscr{N}$ such that, for all bounded continuous functions $h : \mathbb{R}^{n+1} \mapsto \mathbb{R}$, the following conditions hold:

$$|\mathbb{E}\left[h(\boldsymbol{X}, Y); \mathcal{M}\right] - \mathbb{E}\left[h(\boldsymbol{X}, Y); \mathcal{N}\right]| < \epsilon, \qquad \left\|\mathbb{E}_{\boldsymbol{x}}\left[Y; \mathcal{M}\right] - \mathbb{E}_{\boldsymbol{x}}\left[Y; \mathcal{N}\right]\right\|_1 < \epsilon \qquad (2)$$

That is, the subset $\mathcal{N}$ is *dense* in the set of all SCMs $\mathcal{M}$ with regard to (w.r.t.) the $L_1$ norm.

One may surmise that since the parametric forms of simple functions are restrictive (compared to an arbitrary structural function), the family of causal canonical models is not sufficient in representing all the observational distribution (i.e., data) and treatment effects (query) in an arbitrary causal model. Our following result shows that this is not the case.

**Theorem 1.** *The set of CCMs $\mathcal{N}$ is dense in the set of all SCMs $\mathcal{M}$.*

Thm. 1 says that for an arbitrary SCM $\mathcal{M}$, there exists a causal canonical model $\mathcal{N}$ that converges to $\mathcal{M}$ in the observational distribution $P(\boldsymbol{X}, Y)$. For example, let $h$ be the product of indicator functions $h(\boldsymbol{x}, y) = \mathbb{1}_{\boldsymbol{x} \leq \boldsymbol{x}'} \mathbb{1}_{y \leq y'}$ for constants $\boldsymbol{x}' \in \mathbb{R}^n$ and $y \in \mathbb{R}$. Among the equations above, the first term ensures that the cumulative observational distribution $P\left(\boldsymbol{X} \leq \boldsymbol{x}', Y \leq y'\right)$ induced by the canonical model $\mathcal{N}$ converges to the same distribution function in the causal model of ground truth $\mathcal{M}$. In addition, the second term ensures that the canonical model $\mathcal{N}$ could approximate the treatment effects $\mathbb{E}_{\boldsymbol{x}}\left[Y\right]$ for every action $\boldsymbol{x} \in \mathbb{R}^n$ with arbitrary precision with respect to the $L_1$ norm errors.

**Discretizing Action Space.** For the remainder of this section, we will introduce the necessary tools to construct the causal canonical model. First, we will describe a simple function parametrization of the treatment effect that allows us to partition the action space into finite equivalence classes.

**Definition 3.** (Simple Treatment effects) Let $\mathcal{M}$ be an SCM with action $\boldsymbol{X}$ and reward $Y$. The simple treatment effect of action $\boldsymbol{X}$ on reward $Y$ is a function $\widehat{\mathbb{E}}_{\boldsymbol{x}}[Y] : \mathbb{R}^n \mapsto \mathbb{R}$ of the form $\widehat{\mathbb{E}}_{\boldsymbol{x}}[Y] = \sum_{i=1}^N \mathbb{E}_{\boldsymbol{x}_i}\left[Y\right] \mathbb{1}_{\mathbb{X}_i}(\boldsymbol{x})$, where $\mathbb{X}_1, \dots, \mathbb{X}_N$ are disjoint subsets in $\mathbb{R}^n$; and $\boldsymbol{x}_i$ is a realization in $\mathbb{X}_i$ for every $i = 1, \dots, N$.

The next proposition ensures that the simple treatment effect function in Def. 3 can effectively approximate measurable treatment effects for all actions $\boldsymbol{x} \in \mathbb{R}^n$ in an arbitrary causal model.

**Proposition 1** (Equivalence Classes of Action). *Let $\mathcal{M}$ be an SCM with action $\boldsymbol{X}$ and reward $Y$. Then for every $\epsilon > 0$, there exists a simple causal effect $\widehat{\mathbb{E}}_{\boldsymbol{x}}[Y]$ such that $\left\|\mathbb{E}_{\boldsymbol{x}}\left[Y\right] - \widehat{\mathbb{E}}_{\boldsymbol{x}}\left[Y\right]\right\|_1 < \epsilon$. Henceforth, we will consistently refer to the finite sequence of subsets $\mathbb{X}_1, \dots, \mathbb{X}_k$ (or realizations $\boldsymbol{x}_1, \dots, \boldsymbol{x}_k$) associated with $\widehat{\mathbb{E}}_{\boldsymbol{x}}[Y]$ as* equivalence classes of action.

Prop. 1 says that for any causal model $\mathcal{M}$, there must exist a simple treatment effect function that is capable of approximating the original treatment effect in $\mathcal{M}$ with arbitrary precision. Moreover, the simple treatment effect in Def. 3 only takes values of the original treatment effect at a finite set of realized actions $\boldsymbol{x}_1, \dots, \boldsymbol{x}_N$. For any other realization $\boldsymbol{x}$, its treatment effect will be approximated with that of the action $\boldsymbol{x}_i$ such that both $\boldsymbol{x}$ and $\boldsymbol{x}_i$ belong to the same equivalence class $\mathbb{X}_i$. It is thus sufficient to only model the treatment effects induced by finite interventions $\mathrm{do}(\boldsymbol{x}_1), \dots, \mathrm{do}(\boldsymbol{x}_N)$.

**Example 2.** Consider again the causal model $\mathcal{M}$ described in Example 1. Evaluating the treatment effect in $\mathcal{M}$ over actions $x \in [-2, 2]$ gives $\mathbb{E}_x\left[Y\right] = -x^2 + 1$. We will next introduce a simple treatment effect $\widehat{\mathbb{E}}_x[Y]$ to approximate the ground-truth function $\mathbb{E}_x\left[Y\right]$. The idea is that for each integer $k \in \mathbb{Z}$, we will decompose the action domain $[-2, 2]$ into $2^{k+2}$ disjoint intervals. We define $\widehat{\mathbb{E}}_x[Y]$ to be the effect $\mathbb{E}_{x_i}\left[Y\right]$ associated with the action $x_i$ equal to the left endpoint of the subinterval into which the input $x$ falls. Specifically, let $\mathbb{X}_i(k)$ be a subinterval contained in $[-2, 2]$ defined as,

$$\mathbb{X}_i(k) = \left\{ x \in [-2, 2] : -2 + \frac{i-1}{2^k} \leq x < -2 + \frac{i}{2^k} \right\}, \text{ for } i = 1, \dots, 2^{k+2} \qquad (3)$$

The function $\widehat{\mathbb{E}}_x[Y]$ is a linear combination of indicator functions over subintervals $\mathbb{X}_i(k)$ given by, $\widehat{\mathbb{E}}_x\left[Y\right] = -\left(-2 + \frac{i-1}{2^k}\right)^2 + 1$ for $x \in \mathbb{X}_i(k)$, $i = 1, \dots, 2^{k+2}$. Fig. 3a shows a graphical representation of this simple function $\widehat{\mathbb{E}}_x[Y]$ for $k = 2$. It is verifiable that for any $\epsilon > 0$, the $L_1$ norm deviation $\left\|\widehat{\mathbb{E}}_x[Y] - \mathbb{E}_x\left[Y\right]\right\|_1 \leq \epsilon$ when $k \geq \log(16/\epsilon)$. ∎

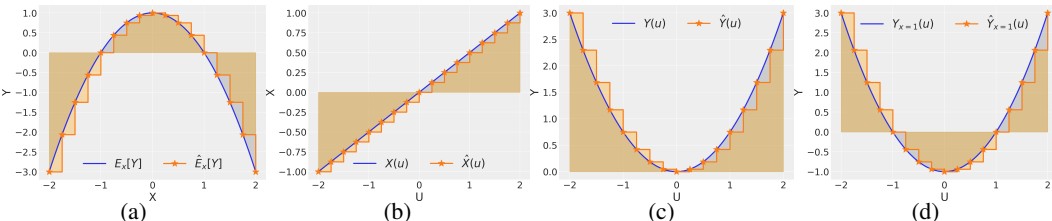

Figure 3: Simple variables to approximate the treatment effect $\mathbb{E}_x[Y]$ (a), observed action $X(u)$ (b) and reward $Y(u)$ (c), and potential outcomes $Y_x(u)$ (d) in the causal model $\mathcal{M}$ of Example 1.

**Discretizing Exogenous Domain.** So far we have decomposed the action domain $\mathbb{R}^n$ and reduced it to a finite set of equivalence classes $\boldsymbol{x}_1, \ldots, \boldsymbol{x}_N$. This means that to effectively approximate the observational distribution and interventional effect in the ground-truth model $\mathcal{M}$, it suffices to model a finite number of potential outcome variables $\boldsymbol{X}, Y$ and $Y_{\boldsymbol{x}_1}, \ldots, Y_{\boldsymbol{x}_N}$. [1] We will next introduce a family of simple functions mapping from the exogenous domains $\mathbb{R}^m$ to the action $\mathbb{R}^n$ or the outcome domain $\mathbb{R}$ to approximate this finite set of potential outcomes.

**Definition 4.** (Simple Potential Outcomes) Let $\mathcal{M}$ be an SCM with action $\boldsymbol{X}$ and reward $Y$. The simple potential outcomes $\widehat{\boldsymbol{X}}, \widehat{Y}$ and $\widehat{Y}_{\boldsymbol{x}}$ are random variables $\widehat{g} : \mathbb{R}^m \mapsto \mathbb{R}$ (or $\widehat{g} : \mathbb{R}^m \mapsto \mathbb{R}^n$) over the exogenous distribution $P(\boldsymbol{U})$ of the form $\widehat{g}(\boldsymbol{u}) = \sum_{i=1}^N g(\boldsymbol{u}_i) \mathbb{1}_{\mathbb{U}_i}(\boldsymbol{u})$, where $\mathbb{U}_1, \ldots, \mathbb{U}_N$ are disjoint subsets in $\mathbb{R}^m$; and $\boldsymbol{u}_i$ is a realization in $\mathbb{U}_i$ for every $i = 1, \ldots, N$.

The following result ensures the effectiveness of simple functions in approximating potential outcomes induced by a finite number of interventions in a ground-truth causal model.

**Proposition 2** (Equivalence Classes of Exogenous). *Let $\mathcal{M}$ be an SCM with action $\boldsymbol{X}$ and reward $Y$. Let $\mathbb{X}$ be a finite set of realizations $\boldsymbol{x} \in \mathbb{R}^n$. Then for every $\epsilon, \delta > 0$, there exist simple potential outcomes $\widehat{\boldsymbol{X}}, \widehat{Y}$ and $\widehat{Y}_{\boldsymbol{x}}$, for all $\boldsymbol{x} \in \mathbb{X}$, that converge to their corresponding potential outcomes $\boldsymbol{X}, Y$ and $Y_{\boldsymbol{x}}$, for all $\boldsymbol{x} \in \mathbb{X}$, almost everywhere. That is,*

$$P\left(\left\{\boldsymbol{u} \in \mathbb{R}^m : \left|\widehat{\boldsymbol{X}}(\boldsymbol{u}) - \boldsymbol{X}(\boldsymbol{u})\right| + \left|\widehat{Y}(\boldsymbol{u}) - Y(\boldsymbol{u})\right| + \sum_{\boldsymbol{x} \in \mathbb{X}} \left|\widehat{Y}_{\boldsymbol{x}}(\boldsymbol{u}) - Y_{\boldsymbol{x}}(\boldsymbol{u})\right| > \epsilon\right\}\right) < \delta, \quad (4)$$

*Henceforth, we will consistently refer to the finite sequence of subsets $\mathbb{U}_1, \ldots, \mathbb{U}_k$ associated with $\widehat{\boldsymbol{X}}, \widehat{Y}$ and $\widehat{Y}_{\boldsymbol{x}}$, for all $\boldsymbol{x} \in \mathbb{X}$, as* equivalence classes of exogenous.

Prop. 2 implies that simple potential outcomes approximate their corresponding potential outcomes in the original causal model almost everwhere. This means that even when realizations of exogenous variables $\boldsymbol{u}$ exist such that $g(\boldsymbol{u}) \neq \hat{g}(\boldsymbol{u})$, the probability assigned to such exogenous realizations must be measure zero. Moreover, the simple potential outcomes only take values of the original variables at a finite number of exogenous realizations $\boldsymbol{u}_1, \ldots, \boldsymbol{u}_N$. It thus suffices to partition the original exogenous domains into a finite number of equivalence classes $\mathbb{U}_1, \ldots, \mathbb{U}_N$, and model the causal mechanisms among observed variables within each equivalence class respectively. Our next example demonstrates the construction of simple potential outcomes.

**Example 3.** Consider the causal model $\mathcal{M}$ described in Example 1. We will introduce simple functions to approximate potential outcomes over a subinterval $[-2, 2]$ in the exogenous domain. For each integer $k \in \mathbb{Z}$, we will decompose the action domain $[-2, 2]$ into $2^{k+2}$ equally sized disjoint intervals. We define $\widehat{X}(u), \widehat{Y}(u)$ and $\widehat{Y}_x(u)$ as the corresponding potential outcomes with unit $u_i$ equal to the left endpoint of the subinterval into which the exogenous value $u$ falls. Specifically, we define $\mathbb{U}_i(k)$ as a subinterval contained in $[-2, 2]$ given by,

$$\mathbb{U}_i(k) = \left\{x \in [-2, 2] : -2 + \frac{i-1}{2^k} \leq u < -2 + \frac{i}{2^k}\right\}, \text{ for } i = 1, \ldots, 2^{k+2} \quad (5)$$

Then simple functions $\widehat{X}(u), \widehat{Y}(u)$ and $\widehat{Y}_x(u)$ are defined as linear combinations of indicator function of subinterval $\mathbb{U}_i(k)$. That is, for $u \in \mathbb{U}_i(k)$, $i = 1, \ldots, 2^{k+2}$, $\widehat{X}(u) = \frac{1}{2}\left(2 - \frac{i-1}{2^k}\right)$, $\widehat{Y}(u) =$

---

[1] We will refer to observed variables $\boldsymbol{X}, Y$ as potential outcomes of the natural intervention $do(\emptyset)$.

$\frac{3}{4}\left(-2 + \frac{i-1}{2^k}\right)^2$, and $\widehat{Y}_x(u) = -x^2 + \left(-2 + \frac{i-1}{2^k}\right)^2$. Fig. 3 shows the graphical description of these simple variables for $k = 2$. Fix an action $x \in [-2, 2]$. It is verifiable that for an arbitrary $\epsilon > 0$, the deviation $\left|\widehat{X}(u) - X(u)\right| + \left|\widehat{Y}(u) - Y(u)\right| + \left|\widehat{Y}_x(u) - Y_x(u)\right| < \epsilon$ when $k \geq \log(9/\epsilon)$. ∎

**Constructing Canonical Models.** We are now ready to put things together and provide a formal justification for the approximation property of canonical models. Specifically, given finite equivalence classes over the action and exogenous domain, one could obtain a canonical model $\mathcal{N}$ that approximates the treatment effect $\mathbb{E}_{\boldsymbol{x}}[Y]$ in the original causal model $\mathcal{M}$ with arbitrary precision. This approximation is supported by Props. 1 and 2.

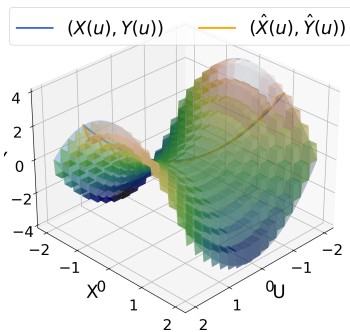

What remains is to ensure that the constructed model $\mathcal{N}$ is consistent with the ground-truth model $\mathcal{M}$ in terms of the observational distribution $P(\boldsymbol{X}, Y)$. For every newly added action $\widehat{\boldsymbol{x}}_i$, we set its simple potential outcome $\widehat{Y}_{\widehat{\boldsymbol{x}}_i'}(\boldsymbol{u}_i)$ as the observed outcome $\widehat{Y}(\boldsymbol{u}_i)$ in the constructed model $\mathcal{N}$; for every unit such that $\widehat{\boldsymbol{X}}(\boldsymbol{u}_j) \neq \widehat{\boldsymbol{x}}_i$, we will find the original action equivalence class $\mathbb{X}_i$ such that $\widehat{\boldsymbol{x}}_i \in \mathbb{X}_i$, and set the simple potential outcome

Figure 4: A simple function approximating the reward function $f_Y(x, u)$ in the ground-truth causal model $\mathcal{M}$ of Example 1.

$\widehat{Y}_{\widehat{\boldsymbol{x}}_i'}(\boldsymbol{u}_j) = \widehat{Y}_{\boldsymbol{x}_i}(\boldsymbol{u}_j)$. Thanks to the construction of simple functions in Defs. 3 and 4, this adjustment ensures the canonical model $\mathcal{N}$'s consistency on the observational distribution $P(\boldsymbol{X}, Y)$ while maintaining its consistency on the treatment effect $\mathbb{E}_{\boldsymbol{x}}[Y]$.

**Example 4.** Consider the causal model $\mathcal{M}$ described in Example 1. We will construct a canonical model $\mathcal{N}$ to approximate the observational distribution $P(X, Y)$ and treatment effects $\mathbb{E}_x[Y], \forall x \in [-2, 2]$. The idea of the construction is to define simple functions $\widehat{f}_X, \widehat{f}_Y$ determining values of action $X$ and outcome $Y$ over the subintervals described in Examples 2 and 3. We then add observed actions to the equivalence class of the action domain to ensure both models are consistent in observational distribution. Specifically, $\widehat{f}_X(u) = \frac{1}{2}\left(2 - \frac{i-1}{2^k}\right) f$, for $u \in \mathbb{U}_i(k), i = 1, \ldots, 2^{k+2}$, and

$$\widehat{f}_Y(x, u) = \begin{cases} \frac{3}{4}\left(-2 + \frac{j-1}{2^k}\right)^2, & \text{if } x = \widehat{f}_X(u) \text{ and } u \in \mathbb{U}_j(k), \ j = 1, \ldots, 2^{k+2} \\ -\left(-2 + \frac{i-1}{2^k}\right)^2 + \left(-2 + \frac{j-1}{2^k}\right)^2, & \text{if } x \in \mathbb{X}_i(k), u \in \mathbb{U}_j(k), \ i, j = 1, \ldots, 2^{k+2} \end{cases}$$

Among the above equations, the first condition in $\widehat{f}_Y(x, u)$ ensures that the canonical model $\mathcal{N}$ induces the simple outcome variable $\widehat{Y}(u)$ described in Example 3. It follows from the definition of $\widehat{X}(u), \widehat{Y}(u)$ that they converge to observed variables $X(u), Y(u)$ in probability. We show in Fig. 4 the graphical representation of the simple reward function $\widehat{f}_Y$ for $k = 5$; the ground-truth reward function $f_Y$ is shown in a blue surface. We also highlight the observed trajectories $(X(u), Y(u))$ in the causal model $\mathcal{M}$ and the canonical model $\mathcal{N}$ in blue and orange respectively. One could see by inspection that the observational distribution $P(\widehat{X}, \widehat{Y})$ in the canonical model $\mathcal{N}$ must converge to the observational distribution $P(X, Y)$ in the causal model $\mathcal{M}$ as $k$ increases. ∎

## 4 GENERATIVE MODELING FOR CAUSAL INFERENCE

For a canonical model in Def. 1, the exogenous domain is discretized, and simple functions decide the values of action and reward. Note that every simple function could be approximated by a continuous function almost everywhere (Stein & Shakarchi, 2009, Ch. 1), leading to further simplification.

**Definition 5.** A causal generative model (CGM) $\mathcal{M}$ is an SCM where exogenous variables $\boldsymbol{U} = \{U\}$ are drawn from a *discrete distribution* $P(U)$ over a finite domain $\{1, \ldots, d\}$. For every unit $U = u$, values of $\boldsymbol{X}$ are drawn from a conditional distribution $P(\boldsymbol{X} \mid u)$ and values $Y$ are decided by a *continuous function* $h_u : \mathbb{R}^n \mapsto \mathbb{R}$ mapping from the action domain $\mathbb{R}^n$ to reward domain $\mathbb{R}$.

Let $\widehat{\mathscr{N}}$ denote the family of all causal generative models compatible with Fig. 2a. It follows immediately from Thm. 1 that the causal approximation property (Def. 2) also holds for CGMs.

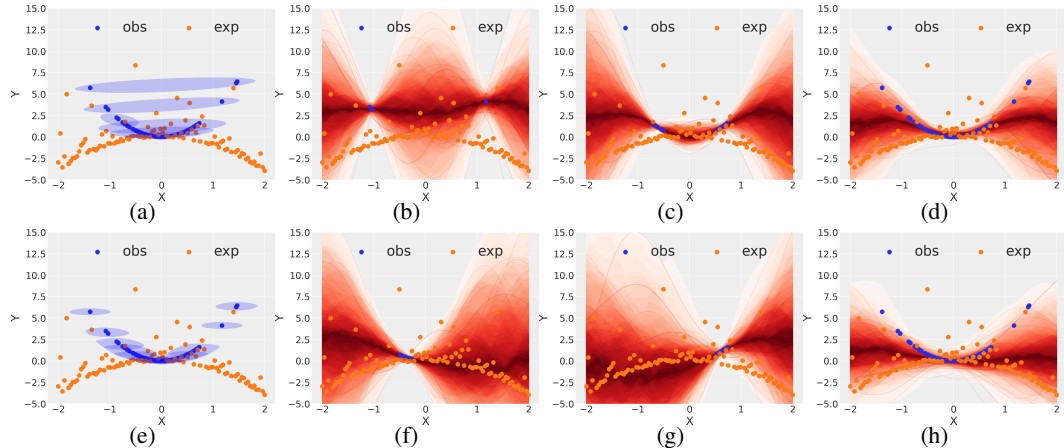

Figure 5: (a) Stratified observed data based on the assigned functional types; (b, c) posteriors over selected canonical functions learned over stratified observations; (d) the posterior over the treatment effect conditioning on observed data; (e - h) alternative stratification over observed data and resulting posteriors over canonical functions and target causal effect conditioning on confounded observations.

**Proposition 3.** *The set of CGM $\bar{\mathscr{N}}$ is dense in the set of all SCMs $\mathscr{M}$.*

Def. 5 describes a novel family for mixture models capable of reproducing the confounded observations and treatment effects in any SCM. Specifically, the underlying population is stratified into a finite number of unknown reward functional types $h_u$ mapping from the domain of $\boldsymbol{X}$ to reward $Y$. For every unit/individual $U = u_i$, the nature samples an observed action $\boldsymbol{x}_i \sim P(\boldsymbol{X} \mid u_i)$, assigns this action to unit/individual $u_i$, and receives a subsequent reward $y_i \leftarrow h_{u_i}(\boldsymbol{x}_i)$. Fix an action $\boldsymbol{x} \in \mathbb{R}^n$. The treatment effect $\mathbb{E}_{\boldsymbol{x}}[Y]$ is obtained by averaging the set of all potential reward functions $h_{u_i}$, weighted by exogenous probabilities $P(u_i)$. Formally, $\mathbb{E}_{\boldsymbol{x}}[Y] = \mathbb{E}_{u \sim P(U)}[h_u(\boldsymbol{x})] = \sum_{u=1}^{d} h_u(\boldsymbol{x})P(u)$.

The following example demonstrates the generative process of CGMs, and how it enables one to obtain a robust treatment effect evaluation from confounded observations.

**Example 5.** Consider the causal model $\mathcal{M}$ described in Example 1. Let $\mathbb{C}_i(k) = \mathbb{U}_i(k) \cup \mathbb{U}_{2^{k+2}-i}(k)$, $i = 1, \ldots, 2^{k+1}$ where $\mathbb{U}_i(k)$ is defined in Eq. (5). It follows from the discretization described in Example 4 that exogenous units $u \in \mathbb{C}_i(k)$ where share the same reward function, i.e., $h_i^{(k)}(x) = -x^2 + \left(-2 + \frac{i-1}{2^k}\right)^2$, for $u \in \mathbb{C}_i(k), i = 1, \ldots, 2^{k+1}$. Fig. 5a shows a partition of the observed data according to the type of associated function $h_i^{(k)}(x)$ for $k = 2$; samples generated by the same reward function are highlighted in the same color.

We next apply standard Gaussian process regression to learn the canonical functions $h_i^{(k)}$ using the corresponding observed data; Figs. 5b and 5c show selected posteriors learned from this process. We also compute the posterior over the treatment effect $\mathbb{E}_x[Y]$ by averaging the learned posteriors over each canonical function $h_i^{(k)}(x)$, weighted by probabilities $P(U \in \mathbb{C}_i(k))$ assigned to each partition. The simulation results show that (1) the learned function $h_i^{(k)}(x)$ picks up behaviors of the ground-truth reward function $f_Y$ in the corresponding partition; (2) the learned posterior over $\mathbb{E}_x[Y]$ generalizes well to internveitonal data sampled from the ground-truth causal model. ∎

In the above example, the membership to the same reward stratum $\mathbb{C}_i(k)$ is a sufficient statistic satisfying the backdoor criterion (Pearl, 1995). This allows one to estimate the treatment effect by adjustment on strata $\mathbb{C}_i(k)$. If the propensity score $P(x|u) > 0$ has full coverage, one could eventually identify the ground-truth effect as the number of observed samples increases. On the other hand, the canonical functional stratification $\mathbb{C}_i(k)$ is generally underdetermined by the confounded observations. For example, Fig. 5e shows the alternative partitioning when one further descritizes $\mathbb{C}_i(k)$ into subsets $\mathbb{U}_i(k)$ and $\mathbb{U}_{2^{k+2}-i}(k)$. In this case, $\mathbb{U}_i(k)$ forms a *potential stratification* that could generate the observed data. Adjustment on this partitioning set leads to a posterior with higher variance for actions $x \in [-1, 1]$, as shown in Fig. 5h, but is still consistent with the actual effect.

**Finding Potential Stratifications.** Given observational data, there generally exist multiple functional partitionings compatible with observations, and each partitioning corresponds to one potential evaluation of treatment effects. To obtain a robust posterior over the target effect, it is thus essential to search over such potential partitionings and compute the treatment effects accordingly. The remainder of this section will describe a Bayesian non-parametric method to carry out this process.

We assume the exogenous probabilities $P(u)$ are drawn from a truncated Dirichlet process, determining the total number of reward functional types and their assigned weights. A mental image for such a distribution follows a stick-breaking process (Sethuraman, 1994) which successively breaks pieces off a unit-length stick with size proportional to random draws from a Beta distribution. Formally, for all $u = 1, \ldots, d-1$, $P(u) = \rho_u \prod_{i=1}^{u-1}(1 - \rho_i)$ where $\rho_u \sim \texttt{Beta}\,(\alpha_u, \beta_{\boldsymbol{u}})$; hyperparameters $\alpha_u, \beta_u > 0$. Finally, we truncate this construction by setting $\rho_u = 1$.

Given any unit $u$, values of action $\boldsymbol{X}$ are drawn from a multivariate normal distribution $P(\boldsymbol{X}|u)$. That is, $\boldsymbol{x} \sim \texttt{Normal}\,(\boldsymbol{\mu}_u, \boldsymbol{\Sigma}_u)$, for every $u = 1, \ldots, d$, where $\boldsymbol{\mu}_u \in \mathbb{R}^n$ is a $n$-dimensional mean vector and $\boldsymbol{\Sigma_u} \in \mathbb{R}^{n \times n}$ is a covariance matrix. As a result, the marginal distribution $P(\boldsymbol{x})$ is defined as a finite mixture of Gaussian distributions. Given any unit $u$, values of outcome $Y$ are decided by the function $h_u(\boldsymbol{x})$, the parameters of which are drawn from a Gaussian process. Formally,

$$\forall u = 1, \ldots, d, \qquad y \leftarrow h_u(\boldsymbol{x}), \qquad h_u(\boldsymbol{x}) \sim \texttt{GP}\,(m_u(\boldsymbol{x}), k_u(\boldsymbol{x}, \boldsymbol{x}')), \qquad (6)$$

where $m_u(\boldsymbol{x}) = \mathbb{E}\,[h_u(\boldsymbol{x})]$ is the expected function value given input $\boldsymbol{x}$, and the covariance function $k_u(\boldsymbol{x}, \boldsymbol{x}')$ represents the correlation between function values at different input points $\boldsymbol{x}$ and $\boldsymbol{x}'$, i.e., $k_u(\boldsymbol{x}, \boldsymbol{x}') = \mathbb{E}\,[(h_u(\boldsymbol{x}) - m_u(\boldsymbol{x}))\,(h_u(\boldsymbol{x}) - m_u(\boldsymbol{x}))]$. In practice, the prior mean function is often set to $m_u(\boldsymbol{x}) = \boldsymbol{0}$ to avoid expensive posterior computations. The function $k_u$ is the kernel of the Gaussian process (Micchelli et al., 2006). One very popular choice is the radial basis function kernel, which is defined as $k_u(\boldsymbol{x}, \boldsymbol{x}') = \sigma_u^2 \exp\left(-\frac{\|\boldsymbol{x} - \boldsymbol{x}'\|^2}{2\lambda_u^2}\right)$. The hyperparameters $\lambda_u$ and $\sigma_u^2$ can be varied to increase or reduce the correlations between points and the variability of the resulting function.

Based on this Bayesian non-parametric model described above, one could then draw samples from the posterior over the target treatment effects $\mathbb{E}_{\boldsymbol{x}}\,[Y]$ given finite observational data $\mathcal{D} = \{(\boldsymbol{x}_i, y_i)\}_{i=1}^N$. There exist general Monte-Carlo Markov Chain algorithms to perform this task, including Hamiltonian Monte Carlo (Duane et al., 1987) and the No-U-Turn Sampler (Hoffman et al., 2014).

## 5   SIMULATIONS AND EXPERIMENTS

We demonstrate our algorithms on synthetic causal models with various reward functions. Overall, simulation results support our findings, and the proposed causal generative model allows us to derive robust posteriors consistent with the actual treatment effects, conditioning on confounded observations. See Sec. B for more details on the simulation setup and additional experiments.

**1. Polynomial Function.** Consider again the causal model described in Example 1 where the reward function is a polynomial function $Y \leftarrow -X^2 + U^2$. We apply our proposed causal generative model and Bayesian non-parametric method to derive posteriors over the treatment effects $\mathbb{E}_x\,[Y]$ for latent cardinalities $d = 10, 20, 30$ respectively. Simulation results, shown in Figs. 6a to 6c, indicate that our proposed method is able to derive a posterior robust against the unobserved confounding bias. The posterior generalizes better to the actual experimental data as the latent cardinality $d$ increases.

**2. Logistic Function.** Consider a causal model where values of action $X$ and reward $Y$ are given by $X \leftarrow E \times U$ and $Y \leftarrow 1/(1 + e^{-X + E \times U})$ respectively; $U$ is an unobserved variable drawn from a standard normal distribution, and $E$ is a uniformly drawn over a binary domain $\{-1, 1\}$. Fig. 7 shows the observational (blue) and experimental data (orange) generated by this model. As expected, the standard Gaussian process regression perfectly fits the confounded observations but fails to generalize to the interventional data collected by randomized experiments.

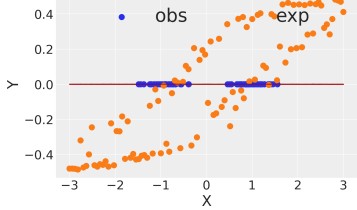

Figure 7: Logistic Function.

We apply our causal method to derive posteriors over the treatment effects $\mathbb{E}_x\,[Y]$ for latent cardinalities $d = 10, 20, 30$ respectively. Results are shown in Figs. 6d to 6f. Our analysis reveals that for $d = 10$, the derived posterior is relatively restrictive, only accounting for interventional data in

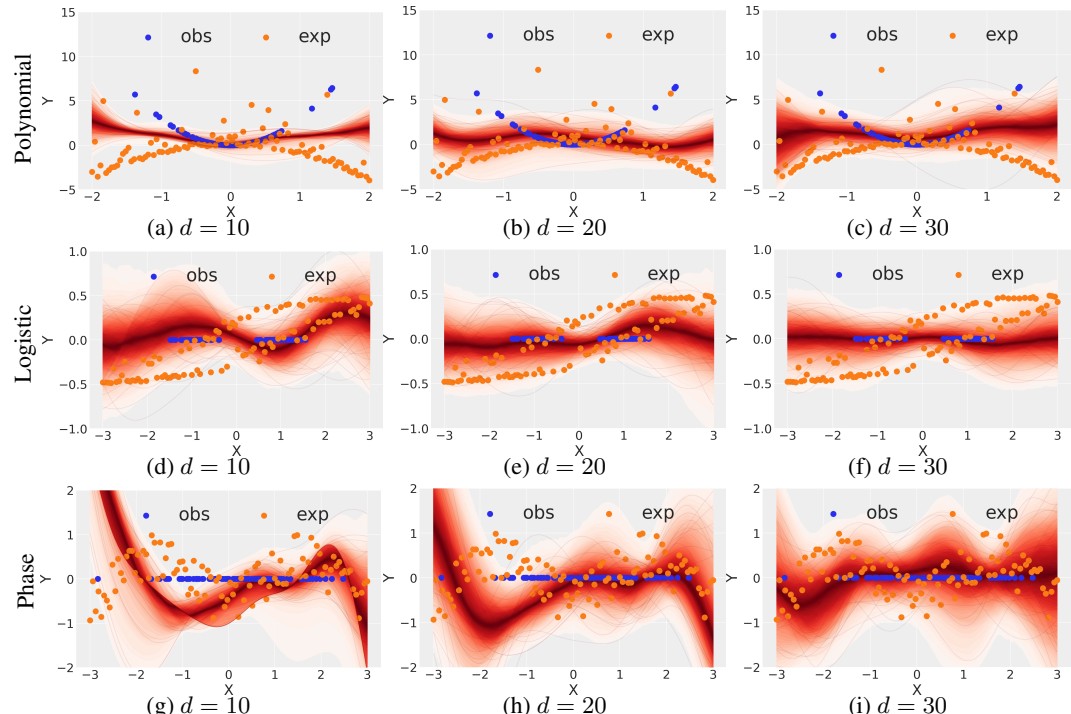

Figure 6: Simulations comparing the derived posteriors over various reward functions using our proposed causal generative model with latent cardinalities $d = 10, 20, 30$ respectively. These functions include: (a - c) polynomial function; (d - f) logistic function; (g - i) phase function.

a single modal of the underlying distribution. As the cardinality $d$ increases (to 30), the resulting generative models can model more complex patterns in the observational and international dynamics, thus leading to a smooth and robust posterior over the treatment effects.

**3. Phase Function.** Consider a causal model where values of action $X$ and reward $Y$ are given by $X \leftarrow U$ and $Y \leftarrow \sin(X)^2 + \cos(U)^2 - 1$ respectively; $U$ is an unobserved variable drawn from a standard normal distribution. We show in Fig. 7 the observational (blue) and experimental data (orange) collected in this model. Like in previous experiments, we applied standard Gaussian process regression to the observational data. The fitted function perfectly models patterns in the observed samples, but fails to generalize to the interventional distribution.

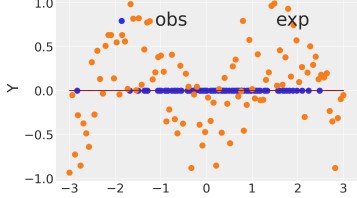

Figure 8: Phase function.

We apply our causal generative model to derive posteriors over the treatment effects $\mathbb{E}_x[Y]$ for latent cardinalities $d = 10, 20, 30$ respectively, and show simulation results in Figs. 6g to 6i. Our proposed method can derive a robust posterior consistent with the actual treatment effects with latent cardinality $d = 30$. Lower cardinalities $d = 10, 20$ fail to generalize due to restricted generative models.

## 6 CONCLUSIONS

This paper addresses the challenge of evaluating causal effects from confounded observational data in complex domains, focusing on the canonical bandit model with basic temporal ordering among action and reward variables. We introduce a new family of causal generative models with a finite number of latent states, which can accurately approximate the observational distribution and treatment effects in any causal model. Using this framework, we present a Bayesian non-parametric method to identify potential stratifications of the observed data, enabling the derivation of posterior distributions over target causal effects despite confounding. Simulation results show that these posteriors generalize well to actual treatment effects and are resilient to unmeasured confounding. Future work will aim to develop models for time series data and integrate additional qualitative causal knowledge.

## REPRODUCIBILITY STATEMENT

The complete proof of all theoretical results presented in this paper, including Thm. 1 and Props. 1 to 3, can be found in Appendix A. Detailed descriptions of the experimental setup are provided in Appendix B. All appendices are included as part of the supplementary material following the "References" section. Please note that all experiments are synthetic and do not involve the introduction of any new assets.

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

## A    PROOFS

This section will provide proof of all the theoretical results in the paper. We will first introduce some propositions that are necessary for the construction of causal canonical models (Props. 1 and 2). We then put things together and formally show the causal approximation property of CCMs (Thm. 1).

**Proposition 1** (Equivalence Classes of Action). *Let $\mathcal{M}$ be an SCM with action $\boldsymbol{X}$ and reward $Y$. Then for every $\epsilon > 0$, there exists a simple causal effect $\widehat{\mathbb{E}}_{\boldsymbol{x}}[Y]$ such that $\left\| \mathbb{E}_{\boldsymbol{x}}[Y] - \widehat{\mathbb{E}}_{\boldsymbol{x}}[Y] \right\|_1 < \epsilon$. Henceforth, we will consistently refer to the finite sequence of subsets $\mathbb{X}_1, \ldots, \mathbb{X}_k$ (or realizations $\boldsymbol{x}_1, \ldots, \boldsymbol{x}_k$) associated with $\widehat{\mathbb{E}}_{\boldsymbol{x}}[Y]$ as* equivalence classes of action.

*Proof.* The proof follows from (Axler, 2020, Prop. 3.44). Specifically, let fix $\mathbb{E}_x[Y]^+$ and $\mathbb{E}_x[Y]^-$ denote the positive and negative parts of the treatment effects $\mathbb{E}_x[Y]$, respectively. For any $\epsilon > 0$ there exist simple functions $\widehat{\mathbb{E}}_{\boldsymbol{x}}[Y]^{(1)}$ and $\widehat{\mathbb{E}}_{\boldsymbol{x}}[Y]^{(2)}$ such that

$$\left\| \mathbb{E}_{\boldsymbol{x}}^+[Y] - \widehat{\mathbb{E}}_{\boldsymbol{x}}^{(1)}[Y] \right\|_1 < \frac{\epsilon}{2}, \qquad \left\| \mathbb{E}_{\boldsymbol{x}}^-[Y] - \widehat{\mathbb{E}}_{\boldsymbol{x}}^{(2)}[Y] \right\|_1 < \frac{\epsilon}{2} \qquad (7)$$

The existence of $\widehat{\mathbb{E}}_{\boldsymbol{x}}[Y]^{(1)}$ and $\widehat{\mathbb{E}}_{\boldsymbol{x}}[Y]^{(2)}$ follows from (Axler, 2020, Prop. 3.9). Let $\widehat{\mathbb{E}}_{\boldsymbol{x}}[Y] = \widehat{\mathbb{E}}_{\boldsymbol{x}}[Y]^{(1)} - \widehat{\mathbb{E}}_{\boldsymbol{x}}[Y]^{(2)}$. Then $\widehat{\mathbb{E}}_{\boldsymbol{x}}[Y]$ is a simple function and

$$\left\| \mathbb{E}_{\boldsymbol{x}}[Y] - \widehat{\mathbb{E}}_{\boldsymbol{x}}[Y] \right\|_1 = \left\| \mathbb{E}_{\boldsymbol{x}}^+[Y] - \widehat{\mathbb{E}}_{\boldsymbol{x}}^{(1)} + \mathbb{E}_{\boldsymbol{x}}^-[Y] - \widehat{\mathbb{E}}_{\boldsymbol{x}}^{(2)}[Y] \right\|_1 \qquad (8)$$

$$\leq \left\| \mathbb{E}_{\boldsymbol{x}}[Y]^+ - \widehat{\mathbb{E}}_{\boldsymbol{x}}[Y]^{(1)} \right\|_1 + \left\| \mathbb{E}_{\boldsymbol{x}}[Y]^- - \widehat{\mathbb{E}}_{\boldsymbol{x}}[Y]^{(2)} \right\|_1 \qquad (9)$$

$$< \epsilon \qquad (10)$$

This proves the statement. $\square$

**Proposition 2** (Equivalence Classes of Exogenous). *Let $\mathcal{M}$ be an SCM with action $\boldsymbol{X}$ and reward $Y$. Let $\mathbb{X}$ be a finite set of realizations $\boldsymbol{x} \in \mathbb{R}^n$. Then for every $\epsilon, \delta > 0$, there exist simple potential outcomes $\widehat{\boldsymbol{X}}, \widehat{Y}$ and $\widehat{Y}_{\boldsymbol{x}}$, for all $\boldsymbol{x} \in \mathbb{X}$, that converge to their corresponding potential outcomes $\boldsymbol{X}, Y$ and $Y_{\boldsymbol{x}}$, for all $\boldsymbol{x} \in \mathbb{X}$, almost everywhere. That is,*

$$P\left( \left\{ \boldsymbol{u} \in \mathbb{R}^m : \left| \widehat{\boldsymbol{X}}(\boldsymbol{u}) - \boldsymbol{X}(\boldsymbol{u}) \right| + \left| \widehat{Y}(\boldsymbol{u}) - Y(\boldsymbol{u}) \right| + \sum_{\boldsymbol{x} \in \mathbb{X}} \left| \widehat{Y}_{\boldsymbol{x}}(\boldsymbol{u}) - Y_{\boldsymbol{x}}(\boldsymbol{u}) \right| > \epsilon \right\} \right) < \delta, \quad (4)$$

*Henceforth, we will consistently refer to the finite sequence of subsets $\mathbb{U}_1, \ldots, \mathbb{U}_k$ associated with $\widehat{\boldsymbol{X}}, \widehat{Y}$ and $\widehat{Y}_{\boldsymbol{x}}$, for all $\boldsymbol{x} \in \mathbb{X}$, as* equivalence classes of exogenous.

*Proof.* We first consider the approximation for a potential outcome variable $Y_{\boldsymbol{x}}(\boldsymbol{u})$. This approximation procedure extends immediately for a combination of observed variables $\boldsymbol{X}(\boldsymbol{u}), Y(\boldsymbol{u})$ and multiple potential outcomes $Y_{\boldsymbol{x}_1}(\boldsymbol{u}), \ldots, Y_{\boldsymbol{x}_k}(\boldsymbol{u})$.

The idea of the proof follows (Axler, 2020, Prop. 2.89). For each $k \in \mathbb{Z}^+$ and $n \in \mathbb{Z}$, the interval $[n, n+1)$ is divided into $2^k$ equally sized half-open subintervals. If $Y_{\boldsymbol{x}}(\boldsymbol{u}) \in [0, k]$, we define $\widehat{Y}_{\boldsymbol{x}}^{(k)}(\boldsymbol{u})$ to be the left endpoint of the subinterval into which $Y_{\boldsymbol{x}}(\boldsymbol{u})$ falls; if $Y_{\boldsymbol{x}}(\boldsymbol{u}) \in [-k, 0)$, we define $\widehat{Y}_{\boldsymbol{x}}^{(k)}(\boldsymbol{u})$ to be the right endpoint of the subinterval into which $Y_{\boldsymbol{x}}(\boldsymbol{u})$ falls; and if $|Y_{\boldsymbol{x}}(\boldsymbol{u})| > k$, we define $\widehat{Y}_{\boldsymbol{x}}^{(k)}(\boldsymbol{u})$ to be $\pm k$. Specifically, let

$$\widehat{Y}_{\boldsymbol{x}}^{(k)}(\boldsymbol{u}) = \begin{cases} \dfrac{m}{2^k} & \text{if } 0 \leq Y_{\boldsymbol{x}}(\boldsymbol{u}) \leq k \text{ and } m \in \mathbb{Z} \text{ is such that } Y_{\boldsymbol{x}}(\boldsymbol{u}) \in \left[ \dfrac{m}{2^k}, \dfrac{m+1}{2^k} \right) \\[2ex] \dfrac{m+1}{2^k} & \text{if } -k \leq Y_{\boldsymbol{x}}(\boldsymbol{u}) \leq 0 \text{ and } m \in \mathbb{Z} \text{ is such that } Y_{\boldsymbol{x}}(\boldsymbol{u}) \in \left[ \dfrac{m}{2^k}, \dfrac{m+1}{2^k} \right) \\[2ex] k & \text{if } Y_{\boldsymbol{x}}(\boldsymbol{u}) > k \\[1ex] -k & \text{if } Y_{\boldsymbol{x}}(\boldsymbol{u}) < -k \end{cases} \quad (11)$$

The definition of $\widehat{Y}_{\boldsymbol{x}}^{(k)}(\boldsymbol{u})$ implies that

$$\left|\widehat{Y}_{\boldsymbol{x}}^{(k)}(\boldsymbol{u}) - Y_{\boldsymbol{x}}(\boldsymbol{u})\right| \leq \frac{1}{2^k} \text{ for all } \boldsymbol{u} \in \mathbb{R}^m \text{ such that } Y_{\boldsymbol{x}}(\boldsymbol{u}) \in [-k, k]. \tag{12}$$

Increase the value of $k \in \mathbb{Z}^+$ such that $1/2^k < \epsilon$ and $P\left(Y_{\boldsymbol{x}}(\boldsymbol{U}) \in [-k, k]\right) < \delta$, which proves the statement for a potential outcome $Y_{\boldsymbol{x}}(\boldsymbol{U})$. We could repeat the same discretization procedure for the observed action $\boldsymbol{X}(\boldsymbol{u})$, observed reward $Y(\boldsymbol{u})$, and every potential outcome $Y_{\boldsymbol{x}}(\boldsymbol{U})$, $\boldsymbol{x} \in \mathbb{X}$. Taking intersections of these partitionings over the exogenous domain $\boldsymbol{U} \in \mathbb{R}^m$ completes the proof. $\quad\square$

**Theorem 1.** *The set of CCMs $\mathcal{N}$ is dense in the set of all SCMs $\mathcal{M}$.*

*Proof.* Given finite equivalence classes over the action and exogenous domain, one could obtain a canonical model $\mathcal{N}$ that approximates the treatment effect $\mathbb{E}_{\boldsymbol{x}}[Y]$ in the original causal model $\mathcal{M}$ with arbitrary precision. Specifically, let $\mathbb{U}^c(k)$ be the set of exogenous values $\boldsymbol{u}$ defined in Prop. 2, and let $\mathbb{U}(k) = \mathbb{R}^m \backslash \mathbb{U}^c(k)$. The $L_1$ error between the treatment effects defined in the SCM $\mathcal{M}$ and the CCM $\mathcal{N}$ could be written as:

$$\|\mathbb{E}_{\boldsymbol{x}}[Y; \mathcal{M}] - \mathbb{E}_{\boldsymbol{x}}[Y; \mathcal{N}]\|_1 = \underbrace{\int_{\mathbb{U}(k)} \|\mathbb{E}_{\boldsymbol{x}}[Y(\boldsymbol{u}); \mathcal{M}] - \mathbb{E}_{\boldsymbol{x}}[Y(\boldsymbol{u}); \mathcal{N}]\|_1 \, dP(\boldsymbol{u})}_{\text{Term 1}} \tag{13}$$

$$+ \underbrace{\int_{\mathbb{U}^c(k)} \|\mathbb{E}_{\boldsymbol{x}}[Y(\boldsymbol{u}); \mathcal{M}] - \mathbb{E}_{\boldsymbol{x}}[Y(\boldsymbol{u}); \mathcal{N}]\|_1 \, dP(\boldsymbol{u})}_{\text{Term 2}} \tag{14}$$

By the construction of set $\mathbb{U}^c(k)$, Term 2 in the above equation gets increasingly smaller as the discretization granularity $k \in \mathbb{Z}^+$ increases. Similarly, one could minimize the $L_1$ error with an arbitrary accuracy by increasing the parameter $k$ and the discretization granularity over the action domain $\boldsymbol{x} \in \mathbb{R}^n$. These approximation properties are supported by Props. 1 and 2.

What remains is to ensure that the constructed model $\mathcal{N}$ is consistent with the ground-truth model $\mathcal{M}$ in terms of the observational distribution $P(\boldsymbol{X}, Y)$. We will achieve this property by enumerating through the finite exogenous equivalence classes $\boldsymbol{u}_1, \dots, \boldsymbol{u}_N$; for every $\boldsymbol{u}_i$, add the approximate observed action $\widehat{\boldsymbol{x}}_i \leftarrow \widehat{\boldsymbol{X}}(\boldsymbol{u}_i)$ to the equivalence classes of action $\mathbb{X}$. For every newly added action $\widehat{\boldsymbol{x}}_i$, we set its simple potential outcome $\widehat{Y}_{\widehat{\boldsymbol{x}}_i'}(\boldsymbol{u}_i)$ as the observed outcome $\widehat{Y}(\boldsymbol{u}_i)$ in the constructed model $\mathcal{N}$; for every unit such that $\widehat{\boldsymbol{X}}(\boldsymbol{u}_j) \neq \widehat{\boldsymbol{x}}_i$, we will find the original action equivalence class $\mathbb{X}_i$ such that $\widehat{\boldsymbol{x}}_i \in \mathbb{X}_i$, and set the simple potential outcome $\widehat{Y}_{\widehat{\boldsymbol{x}}_i'}(\boldsymbol{u}_j) = \widehat{Y}_{\boldsymbol{x}_i}(\boldsymbol{u}_j)$. Since this construction procedure only adds a finite number of points to the equivalence class of the action domain, these points have Lebesgue measure zero and do not affect the $L_1$ approximation error. $\quad\square$

**Proposition 3.** *The set of CGM $\bar{\mathcal{N}}$ is dense in the set of all SCMs $\mathcal{M}$.*

*Proof.* Thm. 1 implies that for any SCM $\mathcal{M} \in \mathcal{M}$, there exists a CCM $\mathcal{N} \in \mathcal{N}$ such that $\mathcal{N}$ approximates the observational distribution and treatment effects in $\mathcal{M}$ with arbitrary accuracy $\epsilon$. It suffices to construct a CGM $\bar{\mathcal{N}} \in \bar{\mathcal{N}}$ approximating the CCM $\mathcal{N}$ with arbitrary accuracy $\epsilon$.

By the definiton of CCM in Def. 1, given any unit $\boldsymbol{u}$, the values of action is given by a constant $\boldsymbol{x}_i$,

$$\widehat{f}_{\boldsymbol{X}}(\boldsymbol{u}) \leftarrow \boldsymbol{x}_i, \tag{15}$$

which could be trivially simulated using a conditional distribution $P(\boldsymbol{X} \mid \boldsymbol{u})$.

Given an unit $\boldsymbol{u}$, values of reward $Y$ in CCM $\mathcal{N}$ are given by a simple function mapping from action domain $\mathbb{R}^n$ to reward domain $\mathbb{R}$. That is,

$$\widehat{f}_Y(\boldsymbol{x}, \boldsymbol{u}) \leftarrow \sum_i^N y_i \mathbb{1}_{\mathbb{X}_i}(\boldsymbol{x}). \tag{16}$$

One could immediately construct a continuous function $h_i : \mathbb{R}^n \mapsto \mathbb{R}$ that approximate the above step function with arbitrary accuracy. The construction follows the same step as (Axler, 2020, Prop. 3.48). This completes the proof. $\quad\square$

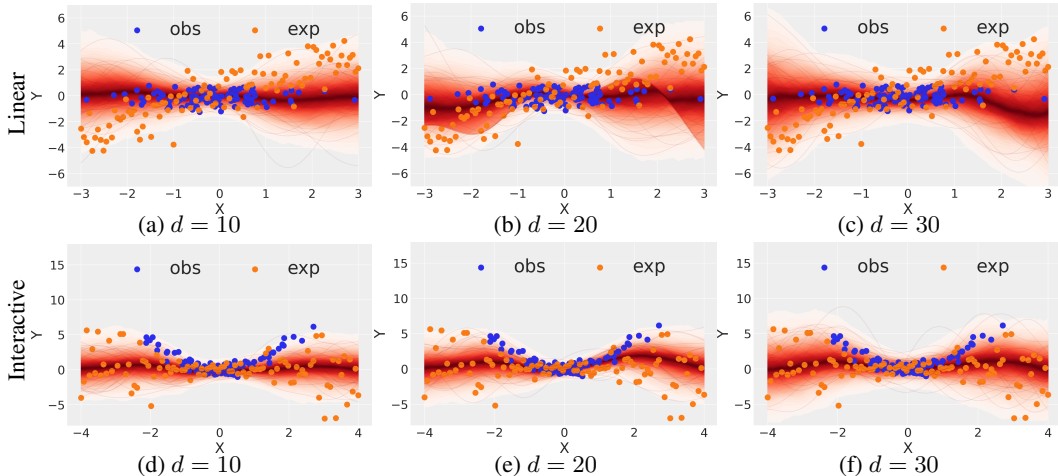

Figure 9: Additional simulations comparing the derived posteriors over various reward functions using the causal generative model with latent cardinalities $d = 10, 20, 30$ respectively. These include: (a - c) linear function; (d - f) a function with interactions between action and latent confounder.

## B    EXPERIMENTAL SETUPS

In this section, we will provide details on the simulation setups. We also conduct additional experiments on other SCM instances. For all experiments, we collect 100 observational samples and use them to draw $1,000$ posterior samples of the treatment effects; each sampled treatment effect function contains potential outcomes $Y_{\boldsymbol{x}}$ valued at 100 different action assignments. All experiments are performed on a Macbook pro laptop with Apple M1 8-core CPU and 32GB memory. We use PyMC (Abril-Pla et al., 2023) as the basic framework for probabilistic programming and inference.

For all experiments, exogenous probabilities $P(u)$ are drawn from a uniform Dirichlet prior `Dirichlet` $(1, \ldots, 1)$. For the treatment assignment distribution $P(\boldsymbol{X}|u)$, the mean $\boldsymbol{\mu}_u$ and variance $\boldsymbol{\Sigma}_u$ are drawn from a standard normal distribution `Normal`$(0, 1)$ and a half normal `Half-Normal`$(0, 0.05)$ respectively. As for the prior distribution over the reward function $h_u(\boldsymbol{x})$, we use the radial basis function kernel with length scale $\lambda_u$ set at constant 1.

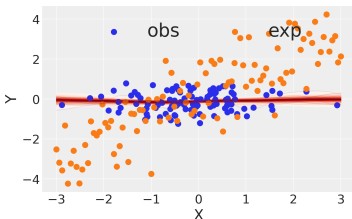

Figure 10: Linear Function.

**4. Linear Function.** Consider a causal model where values of action $X$ and reward $Y$ are given by $X \leftarrow -U$ and $Y \leftarrow X + U + E$ respectively; $U$ is an unobserved variable drawn from a standard normal distribution, and $E$ is an independent noise drawn from a normal distribution `Normal`$(0, 0.5)$. Fig. 10 shows the observational (blue) and experimental data (orange) generated by this model. As expected, the Gaussian process regression perfectly fits the confounded observations but fails to generalize to the interventional data collected by randomized experiments.

We apply our causal method to derive posteriors over the treatment effects $\mathbb{E}_x[Y]$ for latent cardinalities $d = 10, 20, 30$ respectively. Results are shown in Figs. 9a to 9c. Our proposed method can derive a robust posterior consistent with the actual treatment effects with latent cardinality $d \geq 10$.

Figure 11: Interactive function.

**5. Interactive Function.** Consider a causal model where values of action $X$ and reward $Y$ are given by $X \leftarrow U$ and $Y \leftarrow X \times U + E$ respectively; here the reward function contains an interactive term $X \times U$ between the action and the unobserved confounder. $U$ is an unobserved variable drawn from a standard normal distribution, and $E$ is an independent noise drawn from a normal distribution `Normal`$(0, 0.5)$. We show in Fig. 11 the observational (blue) and experimental

data (orange) collected in this model. As expected, standard Gaussian process regression perfectly fits patterns in the observed samples, but fails to generalize to the interventional distribution.

We apply causal generative models to derive posteriors over the treatment effects $\mathbb{E}_x\left[Y\right]$ for latent cardinalities $d = 10, 20, 30$ respectively, and show simulation results in Figs. 9d to 9f. Our proposed method obtains a robust posterior consistent with the actual treatment effects with $d = 30$.

