# OpenReview forum: "Causal Canonical Modeling for Confounding Robust Treatment Evaluation"
_ICLR.cc/2026/Conference — ICLR 2026 Conference Withdrawn Submission_

### Official Review · Reviewer_1F12 · 2025-10-25

**Soundness:** 3
**Presentation:** 3
**Contribution:** 3
**Rating:** 4
**Confidence:** 3

**Summary:**

The paper introduces a new approach for causal inference in the presence of confounding bias, particularly in continuous action-reward systems. It addresses the challenge of estimating treatment effects when there is unobserved confounding, which can lead to spurious correlations in observational data. By using causal canonical models and Gaussian processes, the method allows for the estimation of treatment effects, even when the true causal relationships are non-identifiable.

**Strengths:**

The proposed causal canonical models (CCMs) provide a solid theoretical foundation for approximating causal effects from confounded observational data. The models effectively handle continuous treatment and reward (outcome) variables, making the framework applicable to a wide range of real-world problems, including economics, healthcare, and social sciences.

The paper provides a comprehensive analysis of the causal approximation property of the proposed models and proves that the CCMs are dense in the space of all structural causal models (SCMs).

**Weaknesses:**

The paper does not provide a sufficiently clear exposition of the method used in the simulation study. Including pseudo-code for the entire algorithm would help make the approach more understandable.

The simulation results are not particularly convincing. For instance, in Figure 6a, the posteriors are quite far from the ground truth. Since the paper does not provide rigorous theoretical guarantees for the method, the simulation results alone do not sufficiently convince me that this approach has strong empirical performance.

Following up on the previous comment, the simulation study lacks a benchmark that would demonstrate the performance of the method.

The paper also lacks asymptotic analysis or uncertainty quantification, which would help readers assess the theoretical guarantees. While it is fine to provide simple asymptotic results under strong assumptions, the absence of this analysis makes me feel uncomfortable about the method.

**Questions:**

Have you considered using this framework on real-world datasets? What challenges do you anticipate when applying this method to real-world problems with highly complex data?

Have you thought about targeting a simple parameter instead of the entire treatment response function? For example, how could this method be used to estimate the average treatment effect (ATE)? Additionally, would it be possible to provide simulation results that demonstrate its performance in estimating the ATE?

What are the competing methods in the literature? Can you compare the performance of your method with several competing approaches in the simulation study?

---

### Official Review · Reviewer_XHnU · 2025-10-28

**Soundness:** 1
**Presentation:** 4
**Contribution:** 3
**Rating:** 2
**Confidence:** 3

**Summary:**

The paper studies estimation of causal effects for continuous treatments and outcomes in the presence of an unobserved confounder, a latent state influencing both treatment and outcome. It introduces causal canonical models that discretize the latent state space and define continuous mappings from discrete latent states to continuous treatments and outcomes. The authors show that such a model can approximate observational and interventional distributions to arbitrary accuracy. They further present a practical learning algorithm that models the mapping from discrete latent state to continuous outcome using mixtures of Gaussian processes (GPs), yielding a posterior distribution over the treatment effect. The approach is illustrated on three one‑dimensional simulations.

**Strengths:**

* The paper targets a very important problem in causal inference: estimating treatment effects from observational data. Observational data typically contain unobserved confounding that biases causal estimates; progress on this front could reduce reliance on controlled experiments and impact multiple scientific domains.
* Discretizing the latent state and fitting a mixture of continuous mappings from discrete states to observed treatment and outcome is an original idea. Theoretical results claim that such a model can approximate observational and interventional distributions with arbitrary precision, which is interesting and potentially impactful. The framework appears sound, though I didn’t verify the proofs in the Appendix.
* The paper is well written, notation is clear and the mathematical setup is structured carefully. Figures are high quality, in particular, Figure 1 effectively motivates the problem.

**Weaknesses:**

I’m leaning towards a rejection for two reasons: (i) insufficient empirical support and an unsound evaluation setup, and (ii) lack of consideration of the curse of dimensionality.

**(i) Missing supporting evidence, unsound evaluation setup**

The method reports a posterior distribution over the treatment effect conditional on observed data (e.g., Fig. 5(d,h), Fig. 6). However, no evaluation metrics are provided to assess performance. The manuscript asserts that the learned posterior “generalizes well” to interventional data (e.g., lines 368-369, 419-420, 461-462), but the notion of “goodness” is undefined. As it is, the method functions as a mathematical construct whose empirical goodness cannot be assessed.

The model effectively fits many GPs to small data segments, which are well known to go back to their prior (zero mean with high uncertainty) in the regions where they do not see data. Hence, it should be expected that the posterior of the treatment effect in Example 1 falls back to zero for $|x| > 1$ as there is no data in this region. Indeed, for any SCM satisfying $X := -U/2$ and $Y = 3/4 \times U^2$, the approach would yield the same posterior even when the ground-truth treatment effect samples appear very different, decreasing the reliability of the current qualitative analysis.

**(ii) Curse of dimensionality**

The paper does not discuss how partitioning the latent space, and the associated action/treatment space, scales to high-dimensional actions. Although the theory is stated for actions $X \in \mathbb{R}^n$ of arbitrary $n$, the simulations use only one-dimensional actions. In practice, the number of effective partitions to cover the action space is expected to increase exponentially as $n$ increases. Either the theory should be scoped to the one‑dimensional case, or the manuscript should provide compelling evidence that the method remains effective in high‑dimensional action spaces.

Moreover, contemporary success in representation learning of high-dimensional signals is often attributed to distributed representations that share information about regularities in the data [LeCun+15, Deep Learning]. By contrast, the proposed mixture of GPs entails components that do not share information, which appears to move in the opposite direction.

**Questions:**

The questions about the evaluation metric and scalability to high-dimensional spaces are already stated in the Weaknesses section.

---

### Official Review · Reviewer_A9sn · 2025-10-29

**Soundness:** 3
**Presentation:** 2
**Contribution:** 3
**Rating:** 4
**Confidence:** 2

**Summary:**

This paper tackles the challenge of evaluating causal effects from confounded observational data in continuous action-reward domains. It introduces Causal Canonical Models (CCMs), a novel framework that can universally approximate both observational and interventional distributions of continuous causal models under mild assumptions—temporal ordering and Lebesgue integrability. Building on this theoretical foundation, the authors propose Causal Generative Models (CGMs), formulated as mixtures of Gaussian processes, to infer posterior distributions of unknown treatment effects. The framework enables robust and principled causal inference in settings with unobserved confounding, offering a unified and generalizable approach to policy evaluation beyond the limitations of existing discrete or fully identifiable models.

**Strengths:**

1. This paper addresses an important yet challenging problem in causal inference involving latent confounders.
2. The proposed method is supported by rigorous theoretical analysis.
3. Experimental results demonstrate that the proposed approach.

**Weaknesses:**

1. All experiments are conducted on synthetic SCMs, with no evaluation on real-world observational data (e.g., healthcare or advertising).
2.  The latent cardinality ($d$) must be manually set (e.g., $d=10,20,30$), with no guidance via cross-validation or information criteria, risking underfitting or overfitting in practical applications.
3. The paper does not evaluate CCM/CGM against other representation-learning approaches. Even if those methods incur some confounding bias, the discretization errors in the proposed approach might still lead to worse overall performance.
4. The paper does not propose a method for adaptively discretizing the action and exogenous domains to ensure that the resulting equivalence classes satisfy Proposition 2.

**Questions:**

How would your framework handle cases where part of the covariates are observable? Additionally, when these covariates are high-dimensional, how do you construct equivalence classes, and how do you address potential issues related to the curse of dimensionality?

---

### Official Review · Reviewer_6qAX · 2025-10-29

**Soundness:** 3
**Presentation:** 4
**Contribution:** 2
**Rating:** 4
**Confidence:** 2

**Summary:**

The authors develop a theory of approximating SCMs (with continuous inputs) via simple functions, which they call canonical causal models (CCM). They show that CCMs and their associated simple effects with a fine enough sequence of disjoint subsets approximate causal effects in SCMs. By considering a discretization over the exogenous confounder $U$, the problem of arbitrary unobserved confounding is reduced into the problem of unknown mixture membership. The authors propose a BNP model with a Dirichlet process prior on mixture membership and GP prior on outcome.

**Strengths:**

- The theory about approximating SCMs and causal effects with canonical counterparts is elegant and easy to understand with intuitive examples.

- Replacing continuous unobserved confounding with latent membership is a nice idea to make uncertainty quantification more tractable.

**Weaknesses:**

- It doesn't seem that discretizing the unobserved confounders solves the underlying identification problem. The Dirichlet prior is uninformative and while the posterior may correctly represent uncertainty, I'm not sure that it is able to provide any meaningful insight, beyond a belief about the correct truncation level $d$ (for which a correct choice also doesn't exist in this paper due to the approximation framework). Beyond this the posterior seems to just reflect the underlying non-identifiability, which shows in the Figures presented in the experiments section.

**Questions:**

- Can the posterior of the CGM be used to construct bounds on the treatment effect? Have you tried it on any toy examples?

- Am I right in saying that the posterior here reflects non-identifiability? If so, I think it would be a really nice application of the framework to apply it to partially identified settings (e.g., IV), and to compare it with existing methods. Using the posterior to produce probabilistic bounds/intervals for the causal effect would be an interesting  and elegant perspective on partial identification that the paper currently misses out on.

---

### Official Review · Reviewer_waS9 · 2025-10-31

**Soundness:** 2
**Presentation:** 2
**Contribution:** 1
**Rating:** 2
**Confidence:** 4

**Summary:**

The authors propose a method for Bayesian sampling of the posterior distribution of the average treatment effect and other causal quantities in settings with unobserved confounding. The authors proceed by first establishing that the relevant distributions and average dose response functions can be approximated up to arbitrary precision by linear combinations of simple functions (in the $L_1$ norm), a fact which follows more or less immediately from the usual measure-theoretic construction of the integral. Motivated by these discretization results, the authors propose methods for evaluating the posterior of distribution of causal quantities up to increasing precision. They apply the method to simulated data in order to analyze its performance.

**Strengths:**

Partial identification of causal effects is an important topic. I am sympathetic to the cause of conservative Bayesian inference when the assumptions required for point identification are not credible. I briefly checked the proofs and they all appear to be correct.

**Weaknesses:**

If the causal effect is not identified but only partially identified, then one might wonder what the identified set looks like. If the identified set is large, then the choice of prior will have a very substantial impact on the posterior, even in large samples.  Compared to Zhang et al., (2022) the current paper does not allow additional a priori restrictions on the data-generating process, so that in the present work the identified set is presumably very large if not the entire space.

In the simulations the authors claim that their methods do well to fit the experimental data, but the figures seem to show the opposite. It is not surprising that the posteriors are not concentrated around the true average dose response $E_x[Y]$ given the non-identification: if the identified set is large (as it appears to be in the examples in the paper) then by definition the observed data is uninformative about the causal quantities of interest. What is concerning is that the posteriors do appear to be concentrated, but around functions that are far from the average dose response, which would seem to indicate that the priors are much too concentrated.

The discussion of the approximation by discretization does not seem to me to be very helpful. It is true that by definition any integrable function can be approximated in $L_1$ by simple functions up to arbitrary precision, but these simple functions are linear combinations of characteristic functions over sets that could be very complex. For practical purposes one must instead use simple functions over say, evenly sized intervals, but for that approximation to be effective, one would presumably need to assume that the underlying joint distribution of the outcome, treatment, and confounder, is smooth, which is a distinct exercise from that carried out by the authors. The paper then goes on to use Gaussian process methods, which are justified by noting simple functions can be approximated by continuous ones almost everywhere, but why not just assume continuity in the first place?

I found the paper difficult to follow and confusing in parts. For example, in the section on constructing canonical models the authors discuss the addition of a new action $\hat{x}_i$, I was unclear why actions were being added and to what end. It was only later that the authors explain their sampling process involves first drawing $u_i$ and then some corresponding action, presumably thus constituting such a `new action' $\hat{x}_i$. I also found confusing the discussion that seemed to equate the under-identification with multiple possible stratifications, which starts around the end of page 7.

There is a large literature on Bayesian methods under partial identification, particularly in econometrics (albeit usually in parametric or semiparametric models). That literature considers problems such as prior sensitivity and the frequentist properties of Bayesian credible sets. The authors might find it informative.

The literature review cites many of the same papers repeatedly.

**Questions:**

Could the identified set be characterized in the examples in the paper and if so, what does it look like? How should one go about the problem of choosing the priors?

---

### Note · Authors · 2025-12-03

I have read and agree with the venue's withdrawal policy on behalf of myself and my co-authors.